# Dynamic Scheduled Sampling with Imitation Loss for Neural Text Generation

## Abstract

State-of-the-art neural text generation models are typically trained to maximize the likelihood of each token in the ground-truth sequence conditioned on the previous target tokens. However, during inference, the model needs to make a prediction conditioned on the tokens generated by itself. This train-test discrepancy is referred to as *exposure bias*. Scheduled sampling is a curriculum learning strategy that gradually exposes the model to its own predictions during training to mitigate this bias. Most of the proposed approaches design a scheduler based on training steps, which generally requires careful tuning depending on the training setup. In this work, we introduce **Dy**namic **S**cheduled Sampling with **I**mitation Loss (DYSI), which maintains the schedule based solely on the training time accuracy, while enhancing the curriculum learning by introducing an imitation loss, which attempts to make the behavior of the decoder indistinguishable from the behavior of a teacher-forced decoder. DYSI is universally applicable across training setups with minimal tuning. Extensive experiments and analysis show that DYSI not only achieves notable improvements on standard machine translation benchmarks, but also significantly improves the robustness of other text generation models. Our code is available at: https://anonymous.4open.science/r/DySI-8103.

## 1 Introduction

Advances in deep learning have led to great achievements in neural text generation tasks including machine translation (Vaswani et al., 2017; Wu et al., 2019), summarization (Zhang et al., 2019a; Lewis et al., 2020) and language modeling (Radford et al., 2019; Brown et al., 2020). The dominant approach to date generates the output sequence with a decoder in an autoregressive manner (Bahdanau et al., 2014; Vaswani et al., 2017). To realize the autoregressive formulation, most of the text generation models are trained to maximize the likelihood of each token in the ground-truth sequence conditioned on the previous target tokens with Maximum Likelihood Estimation (MLE). In particular, *Teacher Forcing* (Williams & Zipser, 1989) has been the de facto strategy to help stabilize and speed up the training, where the decoder takes the ground-truth token from the previous time step as the conditioning input for generating the next token. At inference time, however, the decoder does not have access to the previous ground-truth tokens when it is predicting the next token. Thus, the decoder has to instead make a prediction conditioned on the tokens generated by itself so far, resulting in a train-test discrepancy, often referred to as *exposure bias* (Bengio et al., 2015). This discrepancy can lead to error accumulation over time steps as the model might encounter unexpected (though not necessarily wrong) tokens that it has never been exposed to during training.

The methods proposed to combat exposure bias can be categorized into two groups: *Non-MLE-based* approaches (Goyal et al., 2016; Yu et al., 2017; Lin et al., 2017; Nie et al., 2019) and *MLE-based* approaches (Bengio et al., 2015; Song et al., 2021; Liu et al., 2021b). Most non-MLE-based approaches take advantage of generative adversarial networks (GAN) (Goodfellow et al., 2014) and/or reinforcement learning methods to avoid teacher forcing. However, the advantages of these approaches often come with the price of training instability and difficulty, and empirically they still struggle to outperform the MLE baseline (He et al., 2021). On the other hand, MLE-based approaches typically apply curriculum learning (Bengio et al., 2009) strategy to gently bridge the gap between training and inference. These methods often consist of a scheduler, *e.g.,* based on training steps, which controls the extent to which the model should be exposed to its own predictions during training. Intuitively, the model should be exposed to more of its own outputs as the training proceeds.

MLE-based approaches are inherently more efficient and parallelizable as the models do not need to generate the full sequence in inference mode to compute the training loss. Also, MLE has been the mainstream method for training deep neural models. Our work in this paper thus concerns MLE-based training. Bengio et al. (2015) propose *scheduled sampling* to alleviate exposure bias, where the decoder uses the ground-truth previous token as input with probability $\epsilon$, and uses its own prediction with probability $(1 - \epsilon)$. The probability $\epsilon$ is controlled by a scheduler to decay based on the training steps. Such a curriculum learning strategy allows the model to use ground-truth previous tokens at the initial stage of the training and gradually exposes the model to more and more of its own predictions. Zhang et al. (2019b) modify the scheduled sampling of Bengio et al. (2015) by allowing the model to sample from a set of oracle tokens (*e.g.,* synonym of the target token) as the previous token to simulate the model's output at inference time. Goodman et al. (2020) use a stack of $N$ temporal decoders trained to decode along a secondary time axis that allows updating model parameters based on $N$ prediction steps with $N$ being a hyper-parameter. Each decoder gets a predicted previous token as input from its prior decoder, but the training cost increases linearly with $N$. Song et al. (2021) incorporate an error correction mechanism with two decoders. A query stream decoder having access to only positional information first predicts intermediate results, which is then corrected by a content stream decoder. The inference requires running through both decoders, which lowers the decoding efficiency. More recently, Liu et al. (2021b) propose to use a scheduler based on both training and decoding steps. Intuitively, the later decoding steps usually have higher error rates during inference due to error accumulation. Therefore, the model output should be sampled as input with a higher chance for the later decoding steps during training.

As discussed, MLE-based approaches maintain a schedule to decide how much the model should be exposed to its own predictions, which often needs a proxy to estimate the training progress. A schedule (linear or nonlinear) based on training steps usually requires careful design for the specific problem setup as different batch sizes may lead to different training speeds and different tasks may have different convergence rates. This limits the applicability of these approaches to new tasks and datasets. In this work, we introduce **Dy**namic **S**cheduled Sampling with **I**mitation loss (DYSI). First, we propose a scheduler that solely depends on training time accuracy. By tracking training progress though accuracy, we avoid having to perform a costly heuristic search to find a suitable scheduler for each different problem setup. In addition, we use an *imitation loss* to enforce the condition that the generative behavior should match teacher-forced behavior as closely as possible, a core idea in professor forcing (Goyal et al., 2016). Our imitation loss uses the decoder in teacher-forcing mode as the expert to regularize/guide the decoder's behavior when it takes self-generated tokens as input.

We first conduct experiments on machine translation (MT) to demonstrate how our approach performs in various aspects such as generalization and degeneration. Results show that training with DYSI achieves notable improvements on standard MT benchmarks. We then introduce a novel framework for evaluating the robustness of a language model (LM) when exposed to perturbed data, using auto-completion as a test bed. We find that current pre-trained LMs, when trained with standard teacher forcing, are quite sensitive to erroneous contexts that are typical for LM generation such as repetitions. Analysis shows DYSI, by reducing exposure bias, yields a significantly more robust LM across various kinds of perturbations, and overall produces better quality text.

## 2 BACKGROUND

**Text generation.** Typical neural text generation models use an autoregressive factorization of the joint probability over the target sequence. An autoregressive decoder trained with maximum likelihood estimation (MLE) learns to assign a probability to a target sequence $\boldsymbol{y} = (y_1, \cdots, y_T)$ containing $T$ tokens by factorizing the joint probability using the chain rule:

$$\mathcal{L}_{\text{MLE}} = -\sum_{t=1}^{T} \log P(y_t | \boldsymbol{y}_{<t}, \boldsymbol{x}). \tag{1}$$

where $\boldsymbol{x}$ is a source input for conditional text generation (*e.g.,* machine translation) and $\emptyset$ for unconditional generation (*e.g.,* language modeling), and $\boldsymbol{y}_{<t} = (y_1, \ldots, y_{t-1})$ denotes tokens before the current step $t$. To train autoregressive models, teacher forcing (Williams & Zipser, 1989) is commonly used for faster convergence and training stability. In this method, ground-truth tokens from the previous steps are used as input to predict the current token $y_t$. However, it also causes the train-test discrepancy or *exposure bias* as the target tokens are not available at inference time.

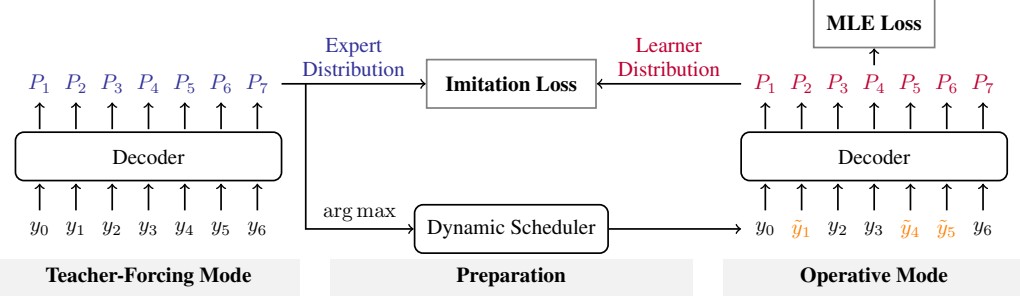

Figure 1: Illustration of Dynamic Scheduled Sampling with Imitation Loss (DYSI). The teacher-forced decoder allows the computation of training accuracy that is directly used in our dynamic scheduler, and it also provides the expert distribution as a supervision signal to the operative decoder. The decoder parameters only get updated when it is in operative mode.

**Scheduled sampling.** Scheduled sampling (Bengio et al., 2015) is a curriculum learning approach to alleviate exposure bias. Instead of conditioning only on the ground-truth context, the model conditions on a sequence $\hat{\boldsymbol{y}}_{<t}$ that mixes tokens from the ground-truth sequence $\boldsymbol{y}_{<t}$ and model's previous predictions $\tilde{\boldsymbol{y}}_{<t}$. Essentially, at each decoding step, a ground-truth token is used as input with a probability of $\epsilon$ and the previous prediction is used with a probability of $(1 - \epsilon)$ as:

$$\hat{y}_t = \begin{cases} y_t & \text{with probability } \epsilon \\ \tilde{y}_t & \text{with probability } (1-\epsilon) \end{cases} \quad (2)$$

The method maintains a decay schedule for the probability $\epsilon$ based on training steps such that the model is exposed to more self-generated tokens at the later stage of the training. The model is trained with the standard MLE loss:

$$\mathcal{L}_{\text{SS}} = -\sum_{t=1}^{T} \log P(y_t | \hat{\boldsymbol{y}}_{<t}, \boldsymbol{x}). \quad (3)$$

## 3 METHODOLOGY

Figure 1 shows an illustration of DYSI. Different from traditional MLE training, it consists of a **dynamic scheduler** and an **imitation** module. The decoder is first run in teacher-forcing mode (henceforth referred to as *teacher-forced decoder*) to obtain an expert distribution, and later in operative mode (henceforth referred to as *operative decoder*) during training. The dynamic scheduler determines the sequence containing target and model-generated tokens that is provided as input to the operative decoder to perform training. The imitation loss constrains the operative decoder behavior to match the teacher-forced decoder's behavior.

### 3.1 DYNAMIC SCHEDULED SAMPLING

Most of the proposed variants of scheduled sampling perform *sampling* differently, *i.e.,* they use different sampling strategies to decide which decoding positions should take the model-generated previous token as input. For example, Bengio et al. (2015); Zhang et al. (2019b) uniformly sample the decoding positions from a sequence with a probability $(1 - \epsilon)$, where $\epsilon$ decays with training steps. Liu et al. (2021a) propose to select the positions where the model has high prediction confidence, *e.g.,* $p(y_t) > 0.9$, while Liu et al. (2021b) propose to sample the positions based on both training and decoding steps with a joint probability distribution function.[1] Instead of proposing a new sampling method, we propose a new **scheduler** that does not rely on training steps, but instead uses the model's performance directly to keep track of the training progress.

Training progress can highly depend on the task, dataset, and experiment setup. For instance, Vaswani et al. (2017) report good performance on WMT'14 En-Fr translation task with $\approx 300$K updates, while Ott et al. (2018b) need only $\approx 90$K updates to get better results with the same model due to a larger batch size. A scheduler based on training steps will inevitably require heuristic-based tuning

---

[1] See Appendix E for approaches that do not adopt sampling strategies based on training steps.

for different experimental conditions, which could be expensive. Moreover, such a scheduler makes the assumption that all the training instances in a batch have the same training progress.

In light of this, we propose to use *training time accuracy* for the schedule, as training accuracy gives more direct feedback about the learning progress. As shown in Figure 1, given a target sequence $\boldsymbol{y} = (y_1, \ldots, y_T)$, we first run the teacher-forced decoder to obtain a sequence of distributions over the vocabulary, $(P_1, \ldots, P_T)$. We then (greedily) sample the distributions to obtain the predictions $\tilde{\boldsymbol{y}} = (\tilde{y}_1, \ldots, \tilde{y}_T)$ where $\tilde{y}_t = \arg\max(P_t) \; \forall t \in [1, \ldots, T]$. We can compute the training time accuracy for a sequence as $Acc(\boldsymbol{y}, \tilde{\boldsymbol{y}}) = (\sum_{t=1}^{T} \mathbb{1}(y_t = \tilde{y}_t))/T$. The scheduler then decides the number of positions ($N$) in the ground-truth sequence to be replaced with tokens generated by the teacher-forced decoder as follows:

$$N \sim \beta \cdot \mathcal{U}(0, Acc(\boldsymbol{y}, \tilde{\boldsymbol{y}}) \cdot T) \tag{4}$$

where $\mathcal{U}$ denotes a uniform distribution and $\beta \in [0, 1]$ is a hyper-parameter that provides further control on the sampling strength in addition to the inherent dynamic control according to training accuracy. Notice that $N$ changes dynamically based on the training accuracy of each instance and is agnostic to training steps. As a sampling strategy, we choose $N$ positions in the sequence randomly and uniformly. A random selection as opposed to selection based on high confidence avoids *confirmation bias* (Tarvainen & Valpola, 2017) where the model accumulates its own errors, and potentially exposes the model to more varied input-output samples, which in turn helps the *behavior cloning*, as discussed in the next section. We view our method, dynamic scheduled sampling and imitation loss, as a unified approach where the the random selection in sampling also contributes to the imitation process.

Ultimately, the output of the dynamic scheduler is a sequence $\hat{\boldsymbol{y}}$ that mixes the tokens generated by the teacher-forced decoder (*i.e.,* tokens from $\tilde{\boldsymbol{y}}$) and the tokens from the ground-truth sequence $\boldsymbol{y}$. This sequence $\hat{\boldsymbol{y}}$ is then used as the input for training the decoder in its operative mode. Note that with a Transformer architecture (Vaswani et al., 2017), all the decoding steps can be trained in parallel (within a layer) by masking out the future tokens.

## 3.2 IMITATION LOSS

In order to mitigate exposure bias entirely, the decoder should behave indistinguishably whether it is fed with a ground-truth or a self-generated token as input. Such a property would allow the model to generalize beyond the context it sees in training. This has been proven effective by Goyal et al. (2016), where they borrow the idea of GAN (Goodfellow et al., 2014) to use a discriminator to distinguish between the hidden states of the decoder in teacher-forcing mode and inference mode. However, putting the decoder in inference mode during training makes the training slow as it requires sampling the full sequence in an autoregressive manner (*i.e.,* no parallelization). Also, training GANs for texts can be challenging as it requires the generator and the discriminator to be on par with each other.

We instead propose to close the gap between the teacher-forced decoder behavior and the operative decoder behavior in the MLE-based setup. To match the operative decoder behavior to the teacher-forced decoder behavior, we bring the intuition of *imitation learning*. In particular, the operative decoder can be seen as a *learner*, which tries to imitate the behavior of an *expert* at each decoding step. This is also known as behaviour cloning.

**Expert.** As shown in Figure 1, the expert, in our case, is the teacher-forced decoder, which provides demonstrations to the learner in the form of sequences of distributions over actions (tokens in the vocabulary). At each decoding step $t$, the expert takes the previous target token $y_{t-1}$ as the observation, and maps its state $s_t$ to the action distribution based on its policy $\pi_{\text{tf}}(s_t) \in \mathbb{R}^{|\mathbb{V}|}$, where $\mathbb{V}$ is the vocabulary and the subscript tf stands for teacher-forcing. More formally,

$$\pi_{\text{tf}}(s_t) = \text{softmax}(s_t) = P_{\theta_{\text{tf}}}(y_t|\boldsymbol{y}_{<t}, \boldsymbol{x}) \tag{5}$$

The expert-generated action distribution is regarded as the supervision signal to guide the learner.

**Learner.** The learner is the decoder running in operative mode in Figure 1. Unlike the expert, it will not always take previous ground-truth tokens as input, instead it will also use the predicted tokens from the expert (see $\arg\max$ in Figure 1) according to the dynamic schedule (§3.1). Specifically, for an observed sequence $\hat{\boldsymbol{y}}$ comprising of both ground-truth and model-generated tokens, the learner

generates an action distribution at every step as:

$$\pi_{\mathrm{op}}(s'_t) = P_{\theta_{\mathrm{op}}}(y_t|\hat{\boldsymbol{y}}_{<t}, \boldsymbol{x}) \tag{6}$$

where op denotes the operative decoder. Notice that the predicted tokens in $\hat{\boldsymbol{y}}$ provide new demonstrations (unseen in the original training data) for the learner comprising states that it may experience during inference. Since the learner and expert share the same parameters, it also simulates the mistakes that the learner may make during inference. Overall, once trained, the operative decoder is expected to behave more robustly under different decoding conditions.

**Learning from expert demonstration.** To match the learner's policy with the expert's, we minimize the Kullback–Leibler divergence (Kullback & Leibler, 1951) ($D_{\mathrm{KL}}$) between the two policies to guide the operative decoder behavior so that it better matches the teacher-forced behavior, considering the latter fixed:

$$\mathcal{L}_{\mathrm{IL}}(\theta_{\mathrm{op}}) = \sum_{t=1}^{T} D_{\mathrm{KL}}(\pi_{\mathrm{tf}}(s_t)||\pi_{\mathrm{op}}(s'_t)) = \sum_{t=1}^{T} D_{\mathrm{KL}}(P_{\theta_{\mathrm{tf}}}(y_t|\boldsymbol{y}_{<t}, \boldsymbol{x})||P_{\theta_{\mathrm{op}}}(y_t|\hat{\boldsymbol{y}}_{<t}, \boldsymbol{x})) \tag{7}$$

Imposing an auxiliary loss to learn from the output distribution of the teacher-forced decoder has another advantage. Although a teacher-forced decoder may fail to predict the exact target token in some positions, it (after being trained enough) often assigns a higher probability mass to the plausible translations in the context, such as synonyms of the target token Li & Lu (2021). Arguably, the soft output distribution contains much more information compared to the one-hot target, which helps the learning of the operative decoder (Furlanello et al., 2018). In Appendix D, we justify why we call the above learning process imitation rather than knowledge distillation (Hinton et al., 2015).

### 3.3 OVERALL TRAINING OBJECTIVE

The generation model is trained with a combination of an MLE loss and the imitation loss:

$$\mathcal{L}(\theta_{\mathrm{op}}) = -\sum_{t=1}^{T} \underbrace{\log P_{\theta_{\mathrm{op}}}(y_t|\hat{\boldsymbol{y}}_{<t}, \boldsymbol{x})}_{\mathrm{MLE}} + \alpha \sum_{t=1}^{T} \underbrace{D_{\mathrm{KL}}(P_{\theta_{\mathrm{tf}}}(y_t|\boldsymbol{y}_{<t}, \boldsymbol{x})||P_{\theta_{\mathrm{op}}}(y_t|\hat{\boldsymbol{y}}_{<t}, \boldsymbol{x}))}_{\mathrm{Imitation}} \tag{8}$$

where $\alpha$ is a hyper-parameter to control the relative weight and $\theta_{\mathrm{tf}} = \theta_{\mathrm{op}}$.

## 4 EXPERIMENTS ON MACHINE TRANSLATION

### 4.1 EXPERIMENTAL SETTINGS

**Datasets & metric.** We evaluate our model on two standard neural machine translation (NMT) benchmarks: WMT'14 English-German (En→De) and English-French (En→Fr). The training datasets contain about 4.5M and 35M sentence pairs respectively. We use `newstest2013` and `newstest2014` as the validation and test sets respectively. The sentences are encoded with joint Byte-Pair Encoding (BPE) (Sennrich et al., 2016) with 40K operations. For performance measure, following previous work, we report the tokenized BLEU (Papineni et al., 2002). We also report other popular translation metrics, *e.g.,* SacreBLEU (Post, 2018), in Table 2. We use the Transformer *big* (Vaswani et al., 2017) as our backbone NMT model (refered to simply as Transformer henceforth). The learning rate warms up to a peak of $0.0008$ with $4,000$ steps, and then decays with the inverse square-root schedule. The value of $\alpha$ in Eq. 8 and $\beta$ in Eq. 4 are set to $0.5$ on both datasets. We use $0.1$ label-smoothing in training and a beam size of $5$ at inference. We train the models with DYSI and teacher forcing for the same number of updates. Appendix A gives further details about the setup.

We also compare against other MLE-based approaches: DynamicConv (Wu et al., 2019), Error Correction (Song et al., 2021), SS-Steps (Liu et al., 2021b) and Scheduled Sampling (Bengio et al., 2015). For SS-Steps, we run their publicly available code using the provided optimized parameters since that produced the best results. For Scheduled Sampling, we adopted the *Exponential* and *Linear* decay schemes from Bengio et al. (2015) and tuned the hyper-parameters based on the validation set performance. Note that the scheduled sampling (Bengio et al., 2015) that was originally proposed for a recurrent architecture samples from the model generated tokens step by step, which is highly inefficient for training large models due to its inability to parallelize. The current paradigm, where

the teacher-forced outputs are sampled as previous outputs, can be seen as an approximation and has been widely used in Transformer-based models (Duckworth et al., 2019; Mihaylova & Martins, 2019; Liu et al., 2021a;b).

## 4.2 TRANSLATION PERFORMANCE

We present the tokenized BLEU scores on WMT `newstest2014` in Table 1. We can observe that our method, *i.e.,* Transformer *big* trained with DYSI achieves 0.9 and 0.5 BLEU improvement on En→De and En→Fr, respectively, over the the same model trained with standard teacher forcing. Our method also outperforms other MLE-based approaches that deal with exposure bias in NMT, such as Scheduled Sampling (SS) and SS-Steps. We also see that training with only Dynamic Scheduled Sampling performs better than or on par with Scheduled Sampling while requiring significantly less tuning, which demonstrates its contribution.

There have been claims that BLEU is not adequate for measuring MT performance and it correlates poorly with human judgements (Chaganty et al., 2018; Post, 2018). In addition to tokenized BLEU, we also report three other popular metrics: deto-kenized SacreBLEU[2](Post, 2018), which tries to solve the problem of difficulty in comparing tokenized BLEU due to differences in tokenization; BLEURT[3] (Sellam et al., 2020), which is a learned evaluation measure based on BERT and is trained with human judgments; COMET[4] (Rei et al., 2020), which leverages cross-lingual pre-trained language modeling and exploits information from both the source and the reference translation

Table 1: Tokenized BLEU scores on `newstest2014` for WMT'14 En→De and En→Fr translation tasks.

| Models | En→De | En→Fr |
|---|---|---|
| Transformer (Ott et al., 2018b) | 29.3 | 43.2 |
| DynamicConv (Wu et al., 2019) | 29.7 | 43.2 |
| Error Correction (Song et al., 2021) | 29.2 | - |
| SS-Steps (Liu et al., 2021b) | 29.6 | 42.8 |
| **Our Implementation** | | |
| Transformer | 29.2 | 42.9 |
| + SS (Bengio et al., 2015) | 29.5 | 43.0 |
| + Dynamic SS | 29.6 | 43.0 |
| + DYSI | **30.1** | **43.4** |

to more accurately predict MT quality. From the results in Table 2, we can observe that our training approach DYSI leads to improvement across all three metrics on both En→De and En→Fr tasks. Our model can outperform the baselines by even larger margins when measured by COMET, which has a high correlation with human judgements.

Table 2: Additional metrics for NMT. ** and * denote significantly better than the Scheduled Sampling with $p < 0.005$ and 0.05, respectively.

| Models | SacreBLEU | BLEURT | COMET |
|---|---|---|---|
| **En-De** | | | |
| Transformer | 28.6 | 58.6 | 35.8 |
| Scheduled Sampling | 28.9 | 58.7 | 36.4 |
| DYSI | 29.4** | 59.1** | 37.3** |
| **En-Fr** | | | |
| Transformer | 41.0 | 54.2 | 59.8 |
| Scheduled Sampling | 41.1 | 54.2 | 60.0 |
| DYSI | 41.4** | 54.5* | 60.7* |

Table 3: Ablation study of $\alpha$ and $\beta$ on WMT'14 En→De development set when the other is set to the default 0.5.

| $\beta$ | 0.0 | 0.25 | 0.5 | 0.75 | 1.0 |
|---|---|---|---|---|---|
| **BLEU** | 26.70 | 27.05 | 27.14 | 27.11 | 27.06 |
| $\alpha$ | 0.0 | 0.25 | 0.5 | 0.75 | 1.0 |
| **BLEU** | 26.84 | 26.97 | 27.14 | 27.06 | 27.09 |

## 4.3 ABLATION STUDY

**Effect of $\beta$ and $\alpha$.** We conduct ablation studies for the two hyper-parameters in DYSI, *i.e.,* $\beta$ in Eq. 4 and $\alpha$ in Eq. 8, to understand how they impact the performance. Intuitively, the larger $\beta$ is, the more positions will be sampled and thus the operative decoder gets to see more of its own predictions during training. Table 3 shows that the performance on the validation set is generally robust to different values of $\beta$ as long as it is larger than a certain value, *e.g.,* 0.25. With a fairly small value of $\beta$, *e.g.,* 0, the model deteriorates to a standard teacher forcing as the scheduler will not sample any positions for using model-generated tokens. On the other hand, $\alpha$ controls the extent to which the operative decoder should imitate the teacher-forced behavior in addition to the original one-hot

---

[2] SacreBLEU signature: `nrefs:1|case:mixed|eff:no|tok:13a|smooth:exp|version:2.0.0`
[3] We use `BLEURT-20-D6`   [4] We use `wmt20-comet-da:xlm-roberta-large`

target. When $\alpha = 0$, DYSI simply becomes another variant of vanilla scheduled sampling. From Table 3, we observe that when $\alpha$ is small, there is a clear gap between model performance and the best result on the validation set, which renders it necessary to include the imitation loss to further boost the performance. We also study two different training initialization strategies in Appendix B.

## 4.4 ANALYSIS

As translation performance alone may not be sufficient to understand how DYSI helps mitigate the exposure bias problem, we conduct further analysis to get more insights.

**Multiple reference translation.** We examine how the model performs against multiple semantically equivalent references. Particularly, we use the dataset from Ott et al. (2018a), which contains 10 additional human translations for 500 sentences from the WMT'14 En→De testset. We present the results in Table 4. Oracle score computes BLEU for every hypothesis *w.r.t.* its best matching reference and averages it over all hypotheses. The oracle scores in Table 4 indicate that the best scores that the models can achieve *w.r.t.* multiple references are comparable. However, the higher corpus score, which measures the BLEU score with all the human references, means that our model has potentially produced more diverse translations, thus having higher coverage over different translations.

Table 4: Corpus BLEU and Oracle Sentence BLEU on WMT14 En→De test set with 10 additional references.

| Model | Single Ref. | Multiple Ref. | |
|---|---|---|---|
| | | Corpus | Oracle |
| Transformer | 28.6 | 74.0 | **83.4** |
| DYSI | **29.4** | **74.8** | **83.4** |

Table 5: Zero-shot translation performance on **WMT'19 Rob**ustness En→Fr task and IWLST'14 En→De test set.

| Model | WMT'19 Rob. | IWSLT |
|---|---|---|
| Transformer | 37.6 | 29.2 |
| DYSI | 38.3 | 29.8 |

We conjecture that DYSI prevents the model from being over-confident and makes the prediction distribution more spread out such that the model tends to use diverse words. To confirm this property, we compute the entropy of the model generation distribution over the vocabulary as $-\sum_{w \in \mathcal{V}} P(y_t = w) \log P(y_t = w)$, and average it over all the decoding steps. The entropy values over the WMT'14 En→De testset for our model and the baseline are 2.22 and 1.79, respectively, confirming our hypothesis.

Previous work (Ott et al., 2018a) points out that excessive spread of the generation distribution may be an indication of over-smoothing, which could happen due to the application of label smoothing. However, unlike the standard label smoothing, where all the classes get smoothed uniformly, the imitation loss in our approach allows the model to learn from the expert distribution through $D_{\mathrm{KL}}$. Learning from a soft target distribution can be seen as an **adaptive** version of label smoothing, which in turn keeps improving for better model regularization and calibration (Müller et al., 2019).

**Robustness.** As mentioned before, exposure bias is closely related to generalization. Wang & Sennrich (2020) also indicate that exposure bias can be more problematic under domain shift. We thus test if DYSI can lead to improvements under a distribution shift. For this, we use the models trained on the WMT'14 dataset (news domain) to perform zero-shot translation on IWSLT'14 En→De testset (Cettolo et al., 2012), consisting of transcribed and translated TED talks (spoken language text). In addition, we use the WMT'19 MT Robustness dataset (Michel & Neubig, 2018) (En→Fr) to investigate how DYSI performs both with a domain shift and non-standard, noisy text. As shown in Table 5, consistent improvements on both tasks indicate that the model trained with DYSI is able to deliver more robust performance compared to the baseline.

## 5 TOWARDS ROBUST TEXT GENERATION

It has been observed that even with large pre-trained language models (LM), high frequency tokens largely dominate generated text (Welleck et al., 2020; Holtzman et al., 2020; Lin et al., 2021). Repetition, both at a single token and at higher $n$-gram levels, is a well known problem in neural text generation. In addition, *oversensitivity* (Jia & Liang, 2017) is a common issue, in which models produce significantly different outputs for very similar inputs, even when the changes preserve semantics (Ribeiro et al., 2018).

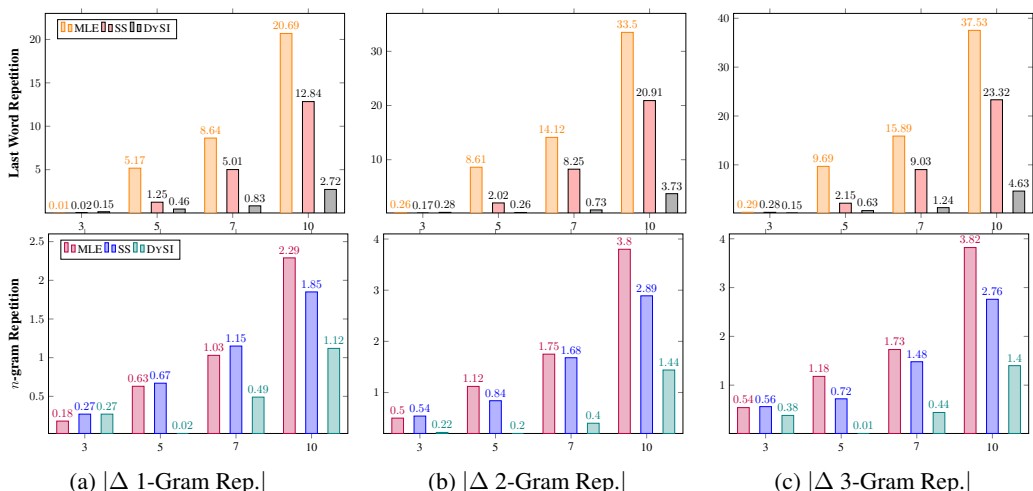

(a) $|\Delta$ 1-Gram Rep.$|$       (b) $|\Delta$ 2-Gram Rep.$|$       (c) $|\Delta$ 3-Gram Rep.$|$

Figure 2: **First row** shows results for Last Word Repetition. The x-axis plots the number of times $m = 3, 5, 7, 10$ that the last word is repeated. **Second row** shows results for $n$-gram Repetition. The x-axis plots the various $n$-gram sizes that we repeat.

Table 6: Examples showing the robustness of GPT-2 trained with DYSI to various perturbations. DYSI is significantly robust to repetition, even in cases of extreme perturbation that causes the baseline to fail irreparably. Text highlighted in red indicates changes due to perturbation, and orange indicates repetition induced by perturbation. (condensed for brevity; best viewed in color).

| Perturbation | Perturbed Prompt | LM Generations |
|---|---|---|
| Last Word Repetition | By 2012 , she was was was was was | **MLE:** was, the Australian National Equestrian Champion , the Australian National Champion
DYSI: to competition were underway, was placed third |
| $n$-gram Repetition | travel with the Doctor starting with "The Bells with "The Bells | **MLE:** and the Doctor starting with " The Bells and the Doctor beginning with " The Bells and the Doctor
DYSI:". It was announced that the third series would be "Clara " |

We conduct experiments to test if training with DYSI can produce a model that is more robust to these kinds of errors. We first fine-tune GPT-2 (Radford et al., 2019) on the WikiText-103 (Merity et al., 2017) training set with MLE loss (standard teacher forcing), Scheduled Sampling (SS) and DYSI. We prompt the trained models with texts extracted from the WikiText-103 test set to perform the auto-completion task. We then perturb these input prompts in an effort to instigate the models to commit repetition errors.

We report MAUVE (Pillutla et al., 2021) scores to compare the machine generated texts with human produced text. MAUVE calculates the area under divergence curves for two text distributions and produces a score[5], where a higher number indicates closer text distributions. MAUVE has been shown to have high correlations with human judgments for open-ended text generation. We use the n-gram repetition ratio difference to evaluate the variations that the perturbations cause. The $n$-gram repetition ratio measures how unique the $n$-grams in a given text are, where a higher score indicates higher repetition and lower diversity. We report the difference between the $n$-gram repetition ratios of two texts for various $n$, which indicates if a given text is more repetitive *w.r.t.* to another. A robust model should produce a diverse and consistent output even when the prompt is perturbed.

## 5.1 Auto-completion

**Fine-tuning LM.** We fine-tune GPT-2 Medium on WikiText-103 (Merity et al., 2017) with a sequence length of 300 tokens and an initial learning rate of 0.0005. Each model is trained for a maximum of 35K iterations and evaluated based on the perplexity on the validation set after every 1K iterations. The perplexity scores on the test set for MLE, SS and DYSI are 13.4, 14.3 and 13.9, respectively.

---

[5] We report scores scaled between 0-100 for readability.

**Prompts for auto-completion.** We use the test set from WikiText-103, which is extracted from a set of verified good and featured articles on Wikipedia. We extract the paragraphs from the document and use the first 50 words as the input prompt to the trained language models. The models need to generate a continuation of 200 BPE tokens based on the prompt. We apply nucleus sampling (Holtzman et al., 2020) ($p = 0.8$) as the decoding strategy since it leads to high-quality and human-like generated text.

**Comparison to human.** We compare the texts generated by the baseline MLE and the DYSI model to the original human continuations using MAUVE. We sample continuation text three times from each model and report the average and standard deviation. MLE achieves a MAUVE score of $71.88 \pm 9.48$, SS achieves a MAUVE score of $72.46 \pm 2.76$, while DYSI achieves a score of $73.08 \pm 3.64$. DYSI has a higher MAUVE score, showing that it produces text that is closer to human-written text. In addition, the baseline MLE model has a significantly higher standard deviation compared to SS and DYSI, showing that models trained with methods that alleviate exposure bias are also more consistent across multiple samplings.

### 5.2 PERTURBATION EXPERIMENTS

We use various strategies to perturb the prompts and compare the $n$-gram repetition ratio differences of model outputs for the perturbed prompts to the model outputs for the original prompts. For these experiments, we sample each model twice for both the original and perturbed prompts, and report the average of all 4 combinations. We also include a comparison of their MAUVE scores in Appendix F.

**Last word repetition.** To test the robustness of the models to repetition, we repeat the last word of the prompt $m = 3, 5, 7, 10$ times, and plot the difference in 1, 2, and 3-gram repetition ratios of the generated text with respect to the text generated for the original prompt. From Figure 2 (row 1), we see that the repetition ratio changes increase significantly with $m$ for the two baseline models. However, DYSI is much more robust. It produces a significantly lower repetition ratio difference.

$n$**-gram repetition.** In this setup, we repeat the last $n$ words of the prompt to test whether DYSI is also robust to repetitions involving longer sequences of words instead of only a single repeated word. We experiment with repeating the last $n = 3, 5, 7, 10$ words and plot the 1, 2, and 3-gram repetition ratio difference of the generated text with respect to the text generated for the original prompt. Interestingly, we see in Figure 2 (row 2) that repeating a longer sequence of words leads to an increase in the repetition ratios for higher order $n$-grams for the MLE baseline. In contrast to both MLE and SS, DYSI maintains a low repetition ratio difference.

Table 6 shows examples of the outputs generated by the baseline MLE and DYSI models for various perturbed prompts. DYSI produces reasonable outputs even with extreme perturbations, and is remarkably robust to repetition perturbations. He et al. (2021) show that LMs trained with teacher forcing have the ability to self-recover. Our experiments demonstrate that DYSI can significantly enhance the model's self-recovery ability from erroneous generation, especially for repetition errors. We also conduct experiments to trigger oversensitivity by perturbing prompts through word replacement. The full experiment setup and results are given in Appendix F. Overall, we find that LMs are generally robust to this, with both SS and DYSI doing better than MLE.

**Experiments on more settings.** We examine if the abovementioned findings are consistent with a **larger model size**. From the results in Appendix G, we see that even with a larger model size (774M), the model trained with the standard teacher forcing is still very sensitive to repetition errors, leading to significant text distribution changes when the repetition error happens. On the other hand, DYSI consistently leads to a more robust language model regardless of the model size. Appendix H further shows the above observations still hold when different **decoding algorithms**, *e.g.,* top-$k$, top-$p$ sampling, are applied.

## 6 CONCLUSION

We have introduced Dynamic Scheduled Sampling with Imitation Loss (DYSI) to combat one of the most well-known problems in autoregressive text generation, exposure bias. It consists of a dynamic sampling scheduler, which keeps track of training progress based on the training accuracy, and an imitation loss, which enables the model to learn from the expert's behavior. DYSI achieves consistent improvement on several translation tasks and experiments. Furthermore, extensive analysis demonstrates that it can yield a significantly more robust text generation model.

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

## A  Hyper-parameters for NMT Experiment

**WMT'14 En→De.**  The learning rate warms up to a peak of $0.0008$ with $4,000$ steps, and decays with the inverse square-root schedule. We apply $0.3$ dropout rate, $0.1$ label smoothing and $0.0$ weight-decay. We use Adam optimizer with $\beta$ being $(0.9, 0.98)$. The training batch size is around 350K tokens. For the random initialization, we train the model for 35K updates. For the hot start initialization, we first the train the model with teacher forcing for 15K updates and use the checkpoint to start retraining (reset the learning rate scheduler) with DYSI or teacher forcing (baseline) for 35K updates. We save the checkpoints every 500 updates and we average the last 10 checkpoints to obtain the final model. During inference, we use a beam size of $5$ and a length normalization factor of $0.2$, which is tuned on the validation set.

**WMT'14 En→Fr.**  The learning rate warms up to a peak of $0.0008$ with $4,000$ steps, and decays with the inverse square-root schedule. We apply $0.1$ dropout rate, $0.1$ label smoothing and $0.0$ weight-decay. We use Adam optimizer with $\beta$ being $(0.9, 0.98)$. The training batch size is around 550K tokens. For the random initialization, we train the model for 50K updates. For the hot start initialization, we first the train the model with teacher forcing for 30K updates and use the checkpoint to start retraining (reset the learning rate scheduler) with DYSI or teacher forcing (baseline) for 50K updates. We save the checkpoints every 500 updates and we average the last 10 checkpoints to obtain the final model. During inference, we use a beam size of $5$ and a length normalization factor of $0.8$, which is tuned on the validation set.

**Computation hardware.**  We conduct the experiment on our machine with CPU Intel(R) Xeon(R) Gold 5218R CPU @ 2.10GHz, and $8\times$ GPU Quadro RTX 6000.

## B  Initialization strategies

**Hot start.** We investigated two strategies to initialize the training with DYSI, namely, random and hot start. In random, all the model parameters are randomly initialized. The other initialization strategy is to first train the model with the standard teacher forcing as it stabilizes training in the initial stage, thus referred to as hot start. Specifically, we first pre-train the model with teacher forcing in the standard setup (§4.1) for 15K and 30K updates on WMT'14 En→De and En→Fr, respectively, and then use the checkpoint to start training with DYSI using the setup described in §4.1.

Table 7: Ablation study of initialization strategies on WMT'14 En→De development set.

| Config | Random | Hot Start |
|---|---|---|
| Transformer | 26.6 | 26.7 |
| DYSI | 27.0 | 27.1 |

Table 7 shows the performance on the WMT'14 En→De validation set for the two strategies. We additionally include the results for the baseline model (teacher forcing) trained with hot start for comparison. Specifically, we take the same pre-trained checkpoint as above for initialization and start training the model with the same training method used to train from scratch. We can see that hot start can generally result in a slightly better performance. We thus adopt the hot start strategy for the NMT experiments. Note that pre-training the models with standard teacher forcing is not exclusive for our approach. In fact, earlier work have adopted the same strategy (Liu et al., 2021a;b).

## C  Performance with varied lengths

We present translation performance with different reference lengths in Table 8. We see that our model is able to consistently outperform the baseline *w.r.t.* all the length buckets. In particular, when the length is longer than 10, the performance gap becomes more significant.

Table 8: Translation performance in BLEU on WMT'14 datasets with varied reference lengths.

| Length | [0,10) | [10,20) | [20,30) | $\geq 30$ |
|---|---|---|---|---|
| Transformer | 22.86 | 28.35 | 29.22 | 29.89 |
| DYSI | 23.22 | 29.33 | 30.18 | 30.42 |
| No. of Sent. | 357 | 1153 | 838 | 655 |

## D  WHY CALL IT IMITATION

**Why call it imitation.** We call the above learning process imitation rather than knowledge distillation or KD (Hinton et al., 2015) for two reasons. First, training an autoregressive generation model can be naturally seen as learning a policy as the model learns to make a sequence of decisions (Pang & He, 2021). Second, KD generally seeks to make a student model learn better from the knowledge extracted from a teacher model (*e.g.,* the prediction distribution) conditioned on the *same input*. However, in our case, the learner may be exposed to a different observation sequence compared to the expert ($\boldsymbol{y}$ vs. $\hat{\boldsymbol{y}}$). In other words, the operative decoder is supposed to imitate the behavior of teacher-forced decoder at each step regardless of the different input.

## E  RELATED WORK

We discuss another line of related work that does not require a scheduler based on training steps. TeaForN (Goodman et al., 2020) uses a stack of $N$ temporal decoders trained to decode along a secondary time axis that allows updating model parameters based on $N$ prediction steps, with $N$ being a hyper-parameter. Each decoder gets a predicted previous token as input from its prior decoder except for the first decoder, but the training cost increases linearly with $N$. Although it does not indeed require curriculum learning, the main difference is that it requires the loss to be backpropagated to all $N$ decoders, while our approach only requires backpropagation to one decoder. SS-Conf (Liu et al., 2021a) decides decoder positions that will receive model outputs based on model confidence, which also does not require sampling based on training steps. However, using a model's confidence is generally not robust as the confidence is closely related to model calibration.

TeaForN achieved $0.1$ and $0.4$ SacreBLEU improvements of on WMT'14 En→De and En→Fr with Transformer Big compared to vanilla Transformer with teacher-forcing. On the other hand, DySI achieved $0.8$ and $0.4$ SacreBLEU improvement on WMT'14 En→De and En→Fr, respectively. As for SS-Conf, we compare to its latest following work SS-Steps (Liu et al., 2021b) in the main paper.

## F  MORE AUTO-COMPLETION RESULTS

We present more results for auto-completion experiment in this section.

**Word replacement.** To trigger oversensitivity in the language models, we randomly replace content words (*i.e.,* nouns, verbs, adjectives and adverbs) in the original prompt with another word. To find a reasonable replacement, we mask the words and use a trained RoBERTa base (Liu et al., 2019) model to generate the replacement word. We experiment with replacing $k = 5, 10, 20$ words and plot the difference in 1, 2, and 3-gram repetition ratios of the generated text with respect to the text generated for the original prompt. We see in Figure 3 that LMs are generally robust to word replacement, with both SS and DYSI doing better than MLE.

**MAUVE scores.** We present the MAUVE scores between the generated texts from different models prompted by the original prompts and the perturbed prompts in Figure 9. We can observe that similar to prior results in 2, the model trained with DYSI is significantly more robust to last word repetition and n-gram repetition problems compared to other baselines.

**Ablation for DYSI.** We provide the ablation results for proposed dynamic scheduled sampling (DS) and imitation loss. In Table 10, the numbers (except PPL) are the average of 1, 2 and 3-gram repetition changes when prompting the trained model with the original and the perturbed text. The best scores between vanilla scheduled sampling (SS) and DS are marked as **bold** and the overall best

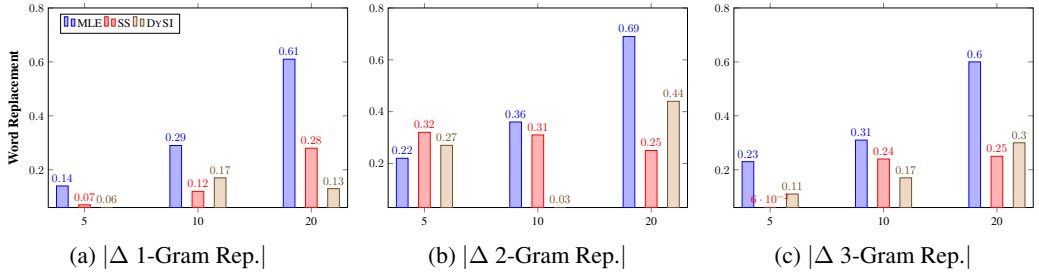

Figure 3: The results for Word Replacement. The x-axis plots the number of words replaced, $k = 5, 10, 20$.

Table 9: MAUVE scores between continuations generated by the same model with the original and perturbed prompts.

| Models | Replacement | | | Last Word Rep. | | | | $n$-gram Rep. | | | |
|--------|------|------|------|------|------|------|------|------|------|------|------|
| | 5 | 10 | 20 | 3 | 5 | 7 | 10 | 3 | 5 | 7 | 10 |
| MLE | 96.20 | 96.54 | 95.08 | 95.72 | 85.71 | 73.45 | 34.76 | 95.58 | 95.87 | 95.56 | 94.65 |
| SS | 96.46 | 96.40 | 96.82 | 96.52 | 95.95 | 85.31 | 57.06 | 96.75 | 95.94 | 95.94 | 95.70 |
| DYSI | 96.49 | 97.09 | 96.54 | 97.20 | 96.69 | 95.13 | 92.56 | 96.77 | 96.87 | 97.25 | 97.01 |

performances are marked with **pink**. Compared to SS, DS has similar overall performance but it is much more robust to last word repetition (also refer to the NMT ablation above). We see that DYSI outperforms others notably in almost all cases due to the imitation loss.

Table 10: Ablation study for auto-completion experiment.

| Models | PPL | Replacement | | | Last Word Rep. | | | | $n$-gram Rep. | | | |
|--------|-----|------|------|------|------|------|------|------|------|------|------|------|
| | | 5 | 10 | 20 | 3 | 5 | 7 | 10 | 3 | 5 | 7 | 10 |
| SS | 14.3 | **0.08** | **0.23** | 0.26 | **0.16** | 1.80 | 7.43 | 19.02 | **0.45** | **0.75** | 1.44 | **2.52** |
| DS | 14.7 | 0.38 | 0.44 | **0.23** | 0.27 | **1.72** | **4.24** | **11.20** | 0.49 | 1.01 | **1.43** | 3.46 |
| MLE | 13.4 | 0.20 | 0.32 | 0.63 | **0.19** | 7.82 | 12.88 | 30.57 | 0.41 | 0.98 | 1.50 | 3.30 |
| DYSI | 13.9 | 0.15 | **0.12** | 0.29 | **0.19** | **0.45** | **0.93** | **3.69** | **0.29** | **0.08** | **0.44** | **1.32** |

**Full examples.** Table 11 shows more examples in auto-completion experiments.

## G  PERTURBATION EXPERIMENTS WITH LARGER MODEL SIZE

We replicate the perturbation experiments with GPT-2 Large (774M). Specifically, we experimented with the perturbation setup of Last Word repetition with $m = 10$ (the last word in the prompt is repeated 10 times), and n-gram repetition with $n = 10$ (last 10 words are repeated). We choose this task to demonstrate that in settings with the maximum perturbation, both the bigger and smaller margin differences between the baseline and DYSI still hold even with GPT2-Large. We sampled the outputs twice from the models trained with standard teacher forcing and DYSI, and report the $n$-gram repetition changes over 4 combinations.

From the results in Table 12, we see that even with a larger size, the model trained with the standard teacher forcing is still very sensitive to repetition errors, leading to significant text distribution changes when the repetition error happens. On the other hand, DYSI consistently leads to a more robust language model regardless of the model size. This experiment shows that DYSI can be effective even with a larger model size, since a larger model size does not necessarily resolve exposure bias.

Table 11: Examples showing the robustness of GPT-2 trained with DYSI to various perturbations. DYSI is significantly robust to repetition, even in cases of extreme perturbation that causes the baseline to fail irreparably. Further, output of the baseline **MLE** training changes completely even when a single token is replaced, but DYSI output maintains the general semantics. Text highlighted in blue indicates original text, red indicates changes due to perturbation, and orange indicates repetition induced by perturbation. (condensed for brevity; best viewed in color).

| Perturbation | Perturbed Prompt | LM Generations |
|---|---|---|
| Last Word Repetition 1 | By 2012 , she was was was was was was was was was was | **MLE:** was, the Australian National Equestrian Champion , the Australian National Champion DYSI: to competition were underway, was placed third |
| Last Word Repetition 2 | He was assigned to its developmental promotion Ohio Valley Wrestling ( OVW ) ) ) ) ) ) | **MLE:** ) ) ) ) ) ) ) ) ) ) ) ) ) ) DYSI: in 2002. Lesnar was involved in the successful submission wrestling a eventually become a member of the WWE |
| $n$-gram Repetition 1 | character that would travel with the Doctor starting with "The Bells with "The Bells | **MLE:** and the Doctor starting with " The Bells and the Doctor beginning with " The Bells and the Doctor DYSI:". It was announced that the third series would be "Clara " |
| $n$-gram Repetition 2 | the Reconnaissance Company particularly distinguished himself . He mounted himself . He mounted | **MLE:** the right flank of the 1st and 2nd Companies and he mounted the right flank of the 3rd and 4th Companies. He mounted his left DYSI: the tank, held it with a pole in the air, and lifted the pole. |
| Word Replacement 1 | After some deliberation Nerva adopted Trajan, a young → wealthy and popular general | **MLE**: given the title of Caesar in 85. → . Nervo attempted to fight DYSI: , as his heir, → , to succeed him as emperor. |
| Word Replacement 2 | reaching 88 K at the crater floor → at the sea level . Under these conditions, | **MLE**: the freezing temperature of the atmosphere → the salt water in the craters forms a "fossil" record DYSI: the shadow temperature is → any significant temperatures would |

Table 12: Results for perturbation experiment with different model sizes.

| | $\Delta$ 1-gram | $\Delta$ 2-gram | $\Delta$ 3-gram |
|---|---|---|---|
| **Last Word Repetition** | | | |
| MLE (Medium) | 20.69 | 33.50 | 37.53 |
| DYSI(Medium) | 2.72 | 3.73 | 4.63 |
| MLE (Large) | 16.21 | 26.20 | 28.99 |
| DYSI(Large) | 2.16 | 3.03 | 3.37 |
| $n$-gran **Repetition** | | | |
| MLE (Medium) | 2.29 | 3.80 | 3.82 |
| DYSI(Medium) | 1.12 | 1.44 | 1.40 |
| MLE (Large) | 2.34 | 3.52 | 3.36 |
| DYSI(Large) | 1.20 | 1.41 | 1.18 |

## H  PERTURBATION EXPERIMENTS WITH DIFFERENT DECODING METHODS

In this section, we conduct experiments on the perturbation task using two of the most popular decoding methods: top-$k$ and top-$p$ sampling with different values of $k$ and $p$. Although we used top-$p$ sampling in our paper with $p = 0.8$, here we show more configurations for comparison. Specifically, we experimented with perturbing the original prompt by repeating the last word 10 times and report the 1, 2, and 3-gram repetition changes (before and after) of the generated text with respect to the text generated for the original prompt. Following the same setup as our main experiment, we sample each model twice for both the original and perturbed prompts to obtain the generated text. The results are averaged over 2 generated texts.

From the results in Table 13, we can observe that the model trained with standard teacher forcing is still sensitive to the repetition errors despite different decoding strategies and different sampling strengths. For example, the generated text has a 3-gram repetition ratio of $0.185$, which is significantly higher than natural text ($0.033$), even with a very strong sampling strength (Top-$p$ with $p = 0.95$). If we compare the outputs with the perturbed prompts to the outputs with the original prompt using the

Table 13: Results for perturbation experiment with different decoding algorithms.

|  | $\Delta$ 1-gram (OP/PP) | $\Delta$ 2-gram (OP/PP) | $\Delta$ 3-gram (OP/PP) |
|---|---|---|---|
| Humans | 0.362 | 0.089 | 0.033 |
| **Top-k** | | | |
| Teacher forcing (k=40) | 0.469/0.529 | 0.166/0.255 | 0.082/0.171 |
| DySI (k=40) | 0.472/0.492 | 0.153/0.163 | 0.068/0.073 |
| Teacher forcing (k=90) | 0.446/0.501 | 0.150/0.230 | 0.073/0.153 |
| DySI (k=90) | 0.442/0.456 | 0.129/0.134 | 0.054/0.058 |
| **Top-p** | | | |
| Teacher forcing (p=0.4) | 0.660/0.878 | 0.444/0.811 | 0.340/0.775 |
| DySI (p=0.4) | 0.649/0.710 | 0.274/0.501 | 0.298/0.040 |
| Teacher forcing (p=0.95) | 0.400/0.486 | 0.120/0.245 | 0.057/0.185 |
| DySI (p=0.95) | 0.392/0.391 | 0.102/0.100 | 0.043/0.045 |

same decoding configuration, we see that DYSI is generally very robust across different decoding strategies and different sampling strengths.

