# OpenReview forum: "Dynamic Scheduled Sampling with Imitation Loss for Neural Text Generation"
_ICLR.cc/2023/Conference — Submitted to ICLR 2023_

### Official Review · Reviewer_8LdF · 2022-10-25

**Confidence:** 4
**Correctness:** 3
**Technical Novelty And Significance:** 3
**Empirical Novelty And Significance:** 3
**Recommendation:** 5

**Clarity, Quality, Novelty And Reproducibility:**

Clarity: good, the paper is clear and easy to understand.

Quality: the design of sampling strategy fair, the experiments are good.

Novelty: the idea that using the signal of training process is kind of novel, the imitation loss is not novel.

Reproducibility: good, the code is provided and should be easy to reproduce.


**Strength And Weaknesses:**

Strength:

- The paper is well written and easy to follow.

- The experiments are comprehensive, and the results are promising

- The idea that using the signal of training process is kind of novel.

Weakness:

- The design of the sampling strategy, especially the Eq. (4), is heuristic. More motivation and analysis of Eq. (4) is required.

- Forwarding a decoder again with modified sequences then computing a loss is not novel because it’s already been used before in similar tasks (e.g., contrastive loss in [1] and consistency loss in [2]). The authors should discuss the differences with them.

[1] BRIO: Bringing Order to Abstractive Summarization, ACL 2022

[2] Target-Side Input Augmentation for Sequence to Sequence Generation, ICLR 2022


**Summary Of The Paper:**

This work presents two techniques to solve the exposure bias. The first one, dynamic scheduled sampling, is to sample the generated token conditioned on the accuracy. The second one, imitation loss, is to regularize the training. The authors conduct experiments on machine translation and robust text generation to evaluate their methods.

**Summary Of The Review:**

Overall, this work tries to solve an important problem of sequence generation and has lots of potential. However, I still have some concerns about the design.

1. Low accuracy comes from two cases: (1) the model is not well trained (e.g., in early stage), (2) the output is actually correct but different from the reference. In case (1) it is reasonable to not sample too much from $\tilde{y}$, but not in the case (2).

2. The accuracy is computed without any alignment. Therefore, we will have a low accuracy when there is just one token deleted/added.

3. The accuracy is only based on single token, not considering the longer sub-sequences such a bi-grams or tri-grams.

4. The accuracy is not considered during sampling. Therefore, when the accuracy is high, there is a high chance that the sampled tokens are identical to reference, making the method less effective.

Therefore, I suggest the authors conduct more theoretical and empirical studies of this part, to make the work more persuasive.

---

### Official Review · Reviewer_tF2J · 2022-10-25

**Confidence:** 4
**Correctness:** 2
**Technical Novelty And Significance:** 2
**Empirical Novelty And Significance:** 1
**Recommendation:** 3

**Clarity, Quality, Novelty And Reproducibility:**

Paper was well written but not reproducible because their link is broken. I don't think adding a KL term on top of scheduled sampling is very novel.

**Strength And Weaknesses:**

This paper does a good job of analyzing why using DySI lets the model produce better, more robust output. They perform the requisite ablations of their method to show it helps. One thing they don't do is check whether repetitions occur with the same frequency as when using regular SS. That is, is their method just a training time optimization (specifically the KL term addition), or does it produce qualitatively different results when you use DySI.

I don't think the NMT results are very impressive. The BLEU scores are < 1 point better, and not even an improvement on the state of the art but rather a Transformer baseline. In my opinion these results are not good enough to publish on.

The abstract has a broken link to the code, so I couldn't review it.

**Summary Of The Paper:**

The authors propose a simple modification of scheduled sampling for transformers that allows them to get better performance on a variety of tasks, including NMT and text generation. Importantly, they analyze why exactly their method does better and produce pretty conclusive quantitative results of this.

**Summary Of The Review:**

Authors apply a minor tweak to scheduled sampling and it works a little better than before.

---

### Official Review · Reviewer_MTWV · 2022-10-25

**Confidence:** 4
**Correctness:** 3
**Technical Novelty And Significance:** 1
**Empirical Novelty And Significance:** 1
**Recommendation:** 3

**Clarity, Quality, Novelty And Reproducibility:**

The paper is clear but lacks novelty. The authors have provided code though I haven't looked at it closely.

**Strength And Weaknesses:**

## Strengths
1. This paper proposes a new decoding training method that combines scheduled sampling with knowledge distillation.
1. The performance looks good and the authors also run a significance test.
1. The method is well-motivated and reasonable.

## Weaknesses
1. Isn't imitation loss knowledge distillation? I see the explanation in Appendix D (probably it's added because it was once asked by reviewers of another venue) -- I'm not convinced by the explanation there. It makes people confused since it hints about the use of reinforcement learning, which is not the case. It simply adds a KD loss to learn from the teacher-forcing distribution.
2. Applying KD to decoding is nothing new. Non-autoregressive decoding uses knowledge distillation to learn from autoregressive decoding. This is very similar to what's proposed in this paper while there isn't a discussion about it.

**Summary Of The Paper:**

This paper proposes Dynamic Scheduled Sampling with Imitation Loss (DySI), which combines knowledge distillation and scheduled sampling for text generation.

**Summary Of The Review:**

This paper applies KD to dynamic schedule sampling but lacks novelty.

---

### Decision · Program_Chairs · 2023-01-20

**Decision:**

Reject

**Justification For Why Not Higher Score:**

The paper is well-written but ultimately proposes an improvement to scheduled sampling that is too similar to previous approaches (knowledge distillation) and lacking motivation and analysis for the new dynamic schedule. The experimental baselines are weak and the results are not strong.

**Justification For Why Not Lower Score:**

N/A

**Metareview: Summary, Strengths And Weaknesses:**

This work proposes a method to fix the train-test discrepancy for autoregressive text generation. It does this by applying scheduled sampling with a schedule based on training time accuracy and also an imitation loss (similar to knowledge distillation).

Strengths:
* The paper is well written.

Weaknesses:
* The imitation loss described seems to be equivalent to knowledge distillation but the explanation is unclear in appendix D.
* Applying knowledge distillation to decoding is not novel and has been done before.
* The experimental results are not convincing and uses a simple Transformer baseline rather than the state of the art.
* The design of the sampling strategy (based on single token accuracy) and heuristic lacks motivation and analysis (such as what happens when the accuracy is high, in which case the method may be less effective).